# Controlled switching thiocarbonylthio end-groups enables interconvertible radical and cationic single-unit monomer insertions and RAFT polymerizations

Wei He [1,3], Wei Tao [2,3], Ze Wei [1], Guoming Tong [1], Xiaojuan Liu [1], Jiajia Tan [2], Sheng Yang [1], Jinming Hu [2], Guhuan Liu [1] ✉ & Ronghua Yang [1] ✉

To emulate the ordered arrangement of monomer units found in natural macromolecules, single-unit monomer insertion (SUMI) have emerged as a potent technique for synthesizing sequence-controlled vinyl polymers. Specifically, numerous applications necessitate vinyl polymers encompassing both radically and cationically polymerizable monomers, posing a formidable challenge due to the distinct thiocarbonylthio end-groups required for efficient control over radical and cationic SUMIs. Herein, we present a breakthrough in the form of interconvertible radical and cationic SUMIs achieved through the manipulation of thiocarbonylthio end-groups. The transition from a trithiocarbonate (for radical SUMI) to a dithiocarbamate (for cationic SUMI) is successfully accomplished via a radical-promoted reaction with bis(thiocarbonyl) disulfide. Conversely, the reverse transformation utilizes the reaction between dithiocarbamate and bistrithiocarbonate disulfide under a cationic mechanism. Employing this strategy, we demonstrate a series of synthetic examples featuring discrete oligomers containing acrylate, maleimide, vinyl ether, and styrene, compositions unattainable through the SUMI of a single mechanism alone.

The endeavor to replicate the ordered arrangement of monomer units along the main chain, akin to natural macromolecules, represents a formidable challenge in polymer synthesis and steers the development trajectory of the next generation of functional materials[1,2]. Vinyl polymers, being the most prevalent and widely utilized polymers, can be synthesized through radical, ionic, or coordination polymerization. The main chain of vinyl polymers contains only stable C-C single bonds, so the properties of the polymer depend on the structure and type of the side chain substituent, that is, the sequence of the monomer units.

The synthesis of sequence-defined vinyl polymers presents challenges due to the intricate process of generating C-C single bonds. Various methods have been reported, including polymerization of oligomers with prearranged monomers[3,4], template polymerization[5,6], chromatographic separation of polydisperse polymers[7,8], atom transfer radical addition (ATRA)[9–11], aminoxyl-mediated single-unit monomer insertion (NM-SUMI)[12], and reversible-addition-fragmentation chain transfer-SUMI (RAFT-SUMI)[12,13]. The development of RAFT-SUMI stemmed from an in-depth exploration of the early stages of RAFT

[1]Key Laboratory of Chemical Biology & Traditional Chinese Medicine Research, Ministry of Education, Institute of Interdisciplinary Studies, College of Chemistry and Chemical Engineering, Hunan Normal University, Changsha 410081 Hunan, China. [2]Department of Polymer Science and Engineering, University of Science and Technology of China, Hefei 230026 Anhui, China. [3]These authors contributed equally: Wei He, Wei Tao. ✉e-mail: ghliu@hunnu.edu.cn; yangrh@pku.edu.cn

polymerization[14], revealing the preferential generation of monoadduct products before further chain growth[15,16]. Subsequent optimization by researchers such as Moad et al.[17–19], Zard et al.[20], Xu et al.[21–24], Boyer et al.[21,22,25], and You et al.[26,27]. extended the application range of RAFT-SUMI, with some utilizing the technique for RAFT step-growth polymerization[26,28,29]. Notably, the reported RAFT-SUMIs predominantly relied on the radical mechanism. In a recent breakthrough, we introduced a RAFT-SUMI based on the cationic mechanism (cSUMI) to broaden the scope of monomers suitable for SUMI[30]. This advancement opens new avenues for expanding the versatility of SUMI and unlocking possibilities in polymer design.

Numerous applications necessitate polymers, especially discrete ones, incorporating both cationically polymerizable monomers like vinyl ethers and radically polymerizable monomers such as styrenes and (meth)acrylates[31–33]. Consequently, a mechanism transformation becomes imperative[31,33–36]. While RAFT-SUMI originates from the RAFT

process, the selection of suitable CTAs for efficient SUMI proves more demanding than that for RAFT. Despite CTAs like xanthate[20,37], dithioester[17], and dithiocarbamate (DTC)[38] offering control over radical RAFT-SUMIs, trithiocarbonates (TTC) with high radical chain transfer constants to a broad monomer range emerge as the optimal choice[18,19], ensuring enhanced mono-adduct yields and broader monomer applicability. Notably, only DTCs have demonstrated efficient monomer addition under cationic RAFT-SUMI[30]. Consequently, the key to achieving the interconversion of radical and cationic SUMI lies in efficiently switching the thiocarbonylthio end group between TTCs and DTCs (Fig. 1).

Drawing on mechanistic analysis, diverse thiocarbonylthio units exhibit interchangeability within the RAFT process. Whitfield et al. and Uchiyama et al. elucidated this phenomenon in the context of radical[39] and cationic[40] RAFTs, respectively. Häkkinen et al.[41] and Courtney et al.[42] utilized TTCs exchange reactions to synthesize heterograft

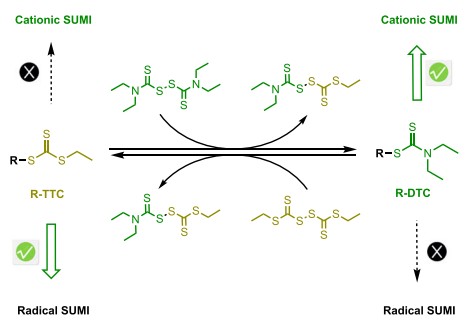 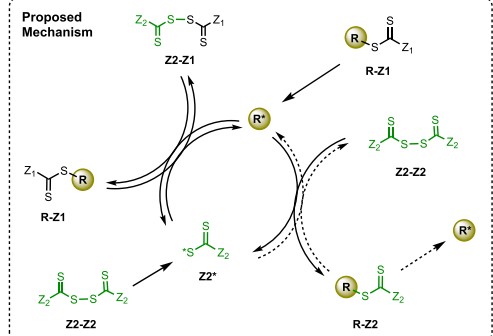

**Fig. 1 | Interconvertible radical and cationic SUMIs and proposed mechanism.** Interconvertible radical and cationic SUMIs by switching thiocarbonylthio end-groups between trithiocarbonates to dithiocarbamates, and proposed mechanism for thiocarbonylthio exchange reaction between chain tranfer agents and bis(-thiocarbonyl) disulfides.

---

## Table 1 | Optimization and control studies for transformation from trithiocarbonates to dithiocarbamates

| No. | R = | Deviation from Standard Conditions[a] | Yield (%)[b] |
|---|---|---|---|
| 1 | -OiBu | None | 92 |
| 2 | -COOEt | None | <5 |
| 3 | -Ph | None | <5 |
| 4 | -CN | None | 8 |
| 5 | -OiBu | Reaction time: 8 h | 36 |
| 6 | -OiBu | Reaction time: 12 h | 74 |
| 7 | -OiBu | Reaction time: 36 h | 91 |
| 8 | -OiBu | 2 eq. TETD | 85 |
| 9 | -OiBu | 5 eq. TETD | 78 |
| 10 | -OiBu | without ZnTPP or Red Light | <5 |
| 11 | -OiBu | Blue Light without ZnTPP | 76 |
| 12 | -OiBu | AIBN at 70 °C | 65 |
| 13 | -OiBu | FcPF$_6$ and DCM | 35 |

[a]Optimizations were performed on 10 mmol scale using TTC. (1 eq.), TETD (1.1 eq.), and ZnTPP (1 mol %) in acetonitrile (MeCN) under red light irradiation (630 nm) over a period of 24 h.
TETD = tetraethylthiuram disulfide, ZnTPP = zinc(II) *meso*-tetraphenylporphine, AIBN = 2,2'-azobis(2-methylpropionitrile), FcPF$_6$ = ferrocenium hexafluorophosphate.
[b]Isolated yield.

copolymers and modify the terminal groups of polymers, respectively. However, the reversibility of thiocarbonylthio exchange lacks controllability, presenting a formidable challenge in achieving a discerning and directed transition from one thiocarbonylthio unit to another.

Herein, we introduced a technique for controllably manipulating thiocarbonylthio end-groups, employing it to transition between cationic and radical RAFT-SUMI. Bis(thiocarbonyl) disulfides, including disulfide bisthiocarbonates, disulfides, and dithiobenzoates, are commonly utilized as intermediates in the synthesis of conventional RAFT CTA[43]. They also play a crucial role in directly governing radical polymerization through their chain transfer properties[44,45]. Consequently, our approach involves the controlled switching of thiocarbonylthio end-groups via an exchange reaction between the chain transfer agent and the bis(thiocarbonyl) disulfides (Fig. 1).

The proposed mechanism of exchange reaction between a CTA (R-Z1) and a bis(thiocarbonyl) disulfide (Z2-Z2) is shown in Fig. 1. The R* species (radical or cation) generated by light/heat cleavage of R-Z1 undergoes the first chain transfer reaction with Z2-Z2 to yield product R-Z2 and Z2* species. Z2* species also yield by directly cleavage of S-S bond within Z2-Z2. The second chain transfer occurs between the Z2* and the R-Z1, regenerating the R* species and releasing the hetero-bis(thiocarbonyl) disulfide, Z1-Z2. The mechanism reveals that if reactant CTA (R-Z1) exhibits a propensity to preferentially yield R-active species over product CTA (R-Z2), the thiocarbonylthio exchange becomes controllable. Consequently, the crux of realizing this concept lies in judiciously selecting appropriate reaction conditions that enable the reactant CTA to preferentially generate specific R-active species.

## Results and discussion

### Transformation from trithiocarbonates to dithiocarbamates

In contrast to chemical methods, photochemistry offers precise control over the selective dissociation of C-S single bonds within thiocarbonylthio compounds. Xu, Boyer, and cowokers reported on the selectivity of photoactivation of thiocarbonylthio compounds for SUMI and selective polymerization through a photoinduced electron/energy transfer (PET) process[46-49]. Their findings revealed that zinc tetraphenylporphine (ZnTPP) could selectively photo-dissociate trithiocarbonates (TTCs) for radical RAFT polymerization but not dithiocarbamates (DTCs). Consequently, this selective photo-dissociation proves instrumental in achieving the conversion from TTCs to DTCs. Under red light, the radical species (R·) generated by the PET process between trithiocarbonate (R-TTC) and ZnTPP facilitates the production of dithiocarbamate (R-DTC) via chain transfer cycles. Since R-DTC cannot regenerate R· under these conditions, the conversion of TTC to DTC proceeds selectively without undergoing the reverse process (Fig. 1).

Our initial attempts involved the reaction of a series of TTCs with distinct R groups (R-TTC) and tetraethylthiuram disulfide (TETD) using ZnTPP under red light irradiation, as detailed in Table 1. Illumination (630 nm) of a solution comprising a vinyl ether-based TTC (R = -OiBu), TETD, and ZnTPP in a 1:1.1:0.01 molar ratio yielded the anticipated product, the corresponding dithiocarbamate (DTC), with an 92% yield (Table 1, entry 1). Additionally, a hetero-bis(thiocarbonyl) disulfide, TTC-DTC, was obtained with an 80% yield (Supplementary Fig. 1). Notably, DTCs derived from vinyl ether-based TTCs proved suitable for subsequent cationic SUMI reactions. However, TTCs featuring conjugated R groups (R = -COOEt, -Ph, and -CN, Table 1, entries 2-4) exhibited considerably lower yields of DTC. This decline in yield was attributed to the inability of the conjugated R· radical, generated through the photo-dissociation of TTC, to add to the C=S double bond within TETD. We proceeded to monitor the reaction process under standard conditions using high performance liquid chromatography (HPLC, Supplementary Fig. 2). The reaction showed an initial induction

period of approximately 2 h, after which it progressed rapidly. Following a 24-h reaction period, we observed that the conversion of iBVE-TTC reached 92% and the HPLC curve did not exhibit significant impurity peaks, except for starting material (iBVE-TTC and TETD). Furthermore, even upon extending the reaction time to 36 h, there was no increase in the conversion of iBVE-TTC. This suggests that the inability to achieve quantitative conversion is attributed to the reversible nature of the reaction. We further purified the reaction systems for 8, 12, and 36 h, yielding 36%, 74%, and 91%, respectively, which closely match their conversion. (Table 1, entries 5-7).

Interestingly, an excess of TETD resulted in a reduction in product yield (Table 1, entries 8 and 9). To explore this phenomenon, we conducted a study on the reaction kinetics under varying ratios of TETD. Our findings indicate that, in contrast to a 1.1-fold of TETD, reactions with a 5-fold of TETD exhibited virtually no induction period, and displayed a significantly faster initial rate, achieving about 42% conversion within 6 h, which is higher than that observed with 2-fold and 1.1-fold excesses of TETD (Supplementary Fig. 3). However, the rate of conversion decreased substantially in the later stages of the reaction. Based on these observations and the proposed mechanism (Supplementary Fig. 4), we hypothesize that under the photocatalysis of ZnTPP, both TETD and iBVE-TTD are capable of generating radicals (DTC radicals and iBVE radicals). A higher concentration of TETD facilitates the rapid initial production of a large quantity of radicals, thereby promoting the TTC-to-DTC transformation and eliminating the induction period. In the later stages of the reaction, the presence of a large excess of TETD results in DTC radicals preferentially reacting

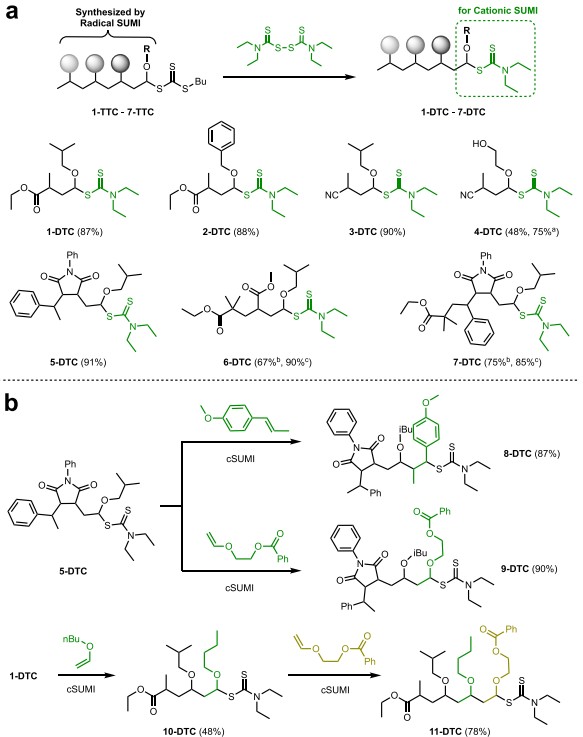

**Fig. 2 | Transformation from trithiocarbonates to dithiocarbamates and further cationic SUMI reaction. a** Transformation from trithiocarbonates (TTCs) to dithiocarbamates (DTCs) via exchange reaction with tetraethylthiuram disulfide (TETD). Optimizations were performed scale using TTC (1-TTC – 7-TTC, 1 eq.), TETD (1.1 eq.), and ZnTPP (1 mol %) in acetonitrile under red light irradiation (630 nm) over a period of 24 h. [a] 1 mol % AIBN was used. [b] Reaction time was extended to 48 h. [c] 2 eq. of TETD was used. **b** Following cationic SUMI of vinyl ethers or styrenes. ZnTPP = zinc(II) *meso*-tetraphenylporphine, AIBN = 2,2′-azobis(2-methylpropionitrile).

with TETD rather than with iBVE-TTC, leading to a slowed rate of product conversion.

In the absence of ZnTPP or red light, the reaction failed to proceed (Table 1, entry 10). Without ZnTPP, blue light irradiation (460 nm) directly initiated the reaction that involved homolytic cleavage of C-S bonds. This led to a relative decrease in DTC yield (76%, Table 1, entry 11). The investigation into the reaction procedure revealed that extending the reaction duration did not enhance the conversion rate (Supplementary Fig. 5). Therefore, the lower yield can be attributed to the fact that blue light also mediates the breakdown of DTC and triggers the reverse transformation of DTC to TTC. Introducing a conventional radical source led to a less selective C-S single bond cleavage, resulting in a 65% yield of DTC in the 2,2′-azobis(2-methylpropionitrile) (AIBN)-initiated system (Table 1, entry 12). As a control, we explored thiocarbonylthio exchange reactions via a cationic initiation system. The carbenium ion (R⁺) generated through the selective oxidation of TTC by ferrocenium salts (Fc⁺)[50] led to a 35% yield of DTC (Table 1, entry 13).

With the optimized system, we expanded our exploration to the scope of TTCs synthesized through continuous radical SUMI (Fig. 2a). TTCs (1-TTC to 7-TTC) were prepared through one-step or successive radical SUMI of styrene, acrylate, maleimide, and vinyl ether into trithiocarbonate CTA, as literature methods with slight modifications (Supplementary Figs. 6–18). Treating these TTCs with TETD yielded the corresponding DTCs (1-DTC to 7-DTC) with good yields ranging from 75% to 91% (Fig. 2a and Supplementary Figs. 19–25). It's noteworthy that due to minimal alterations in the product's polarity during the end-group transformation process, employing flash chromatography, like the commercial chromatography Biotage, results in improved yields. To prepare 4-DTC with a hydroxyl group, the AIBN initiation system is used to achieve a 75% yield, as the presence of the hydroxyl group may react with intermediates in the photoredox process[51–53]. Compounds 6-DTC and 7-DTC have lower reaction rates under standard conditions, with yields of 67% and 75% after 48 h,

respectively. In our investigation of the reaction kinetics between 7-TTC and TETD under standard conditions, we observed a notably induction period of ~6 h. Following this induction period, the reaction proceeded rapidly, reaching an ~80% conversion rate at 36 h and ~93% at 72 h (Supplementary Fig. 26). However, when the amount of TETD was increased to a 2-fold molar excess, the induction period was eliminated, with the reaction achieving about ~87% conversion within 24 h. Extending the reaction time beyond this point did not further increase the conversion rate (Supplementary Fig. 26). Based on the observed kinetics of the reaction, to enhance the synthetic efficiency, we employed a 2-fold of TETD for the synthesis of 6-DTC and 7-DTC, achieving favorable yields.

The DTCs resulting from the exchange reaction between TTC and TETD can serve as chain transfer agents to facilitate subsequent cationic SUMI (Fig. 2b) for appropriate monomers. Recent findings revealed that an electron-deficient vinyl ether or styrene can efficiently insert into a DTC derived from an electron-rich vinyl ether. In anhydrous DCM, a stoichiometric mixture of electron-deficient vinyl ether (2-(vinyloxy)ethyl benzoate) and DTC (5-DTC) reacted with FcPF₆, resulting in mono-adducts 9-DTC with yields of 90%. Subsequent exploration involved the cationic SUMI of alkoxy styrene (trans-anethole) into 5-DTC, yielding mono-adduct 8-DTC with an 87% yield. Additionally, the synthesis of heterotrimer (11-DTC) was accomplished by introducing ether (2-(vinyloxy)ethyl benzoate) into 10-DTC, yielding a 78% yield (Supplementary Figs. 27–30). These outcomes highlight the realization of a transformation in the mechanism from radical SUMI to cationic SUMI, successfully enabling the synthesis of discrete oligomers containing styrene, acrylate, maleimide, vinyl ether, and acrynitrile units.

## Transformation from dithiocarbamates to trithiocarbonates
Utilizing a PET-mediated radical process, TTCs featuring pre-terminal vinyl-ether derivatives were effectively transformed into DTCs through the addition of 1.1 eq. TETD. This successful conversion marked the

**Table 2 | Optimization and control studies for transformation from dithiocarbamates to trithiocarbonates**

| No. | R = | Deviation from standard conditions[a] | Yield (%)[b] |
|-----|-----|---------------------------------------|--------------|
| 1 | -PhOMe | None | 90 |
| 2 | -Ph | None | 8 |
| 3 | -COOEt/-CN | None | <5 |
| 4 | -OiBu | None | 23 |
| 5 | -PhOMe | Reaction time: 6 h | 52 |
| 6 | -PhOMe | Reaction time:12 h | 75 |
| 7 | -PhOMe | Reaction time: 36 h | 89 |
| 8 | -PhOMe | 2 eq. BBTD | 70 |
| 9 | -PhOMe | 1 eq. BBTD | 46 |
| 10 | -PhOMe | without FcPF₆ | 0 |
| 11 | -PhOMe | TMPP and Blue Light | 60 |
| 12 | -PhOMe | ZnTPP and Red Light | <5 |
| 13 | -PhOMe | UV Light (365 nm) | 35 |

[a]Optimizations were performed on 10 mmol scale using DTC (1 eq.), BBTD (5 eq.), FcPF₆ (1 mol %) in DCM over a period of 24 h. FcPF₆ = ferrocenium hexafluorophosphate, BBTD = bis(butylsulfanyl-thiocarbonyl) disulfide, TMPP = 2,4,6-tris(p-methoxyphenyl) pyrylium tetrafluoroborate.
[b]Isolated yield.

achievement of a radical-to-cationic SUMI transformation. The high selectivity of the forward reaction, converting TTC to DTC, made the reverse transformation via the radical process challenging to complete. Building on observations by Kamigaito et al. [40], who noted that under zinc chloride treatment, DTC and xanthate could exchange thiocarbonylthio moieties through a cationic process, we proceeded to explore the conversion of DTCs to TTCs via a cationic-initiated exchange reaction using bis(sulfanyl-thiocarbonyl) disulfide.

The carbocation (R$^+$) generated through the oxidation of dithiocarbamate (R-DTC) by ferrocenium salts (Fc$^+$)[50] initiated chain transfer cycles, ultimately producing trithiocarbonate (R-TTC). The more efficient generation of carbocations by DTCs with electron-donating nitrogen atoms, compared to TTCs under these conditions, favored the forward DTC-to-TTC conversion over the reverse reaction. However, it is crucial to note that TTCs can also undergo oxidation by Fc$^+$ to generate carbocations, introducing the possibility of the reverse reaction. To address this challenge, we plan to enhance the yield by increasing the stoichiometric quantity of bis(sulfuryl-thiocarbonyl) disulfides (Fig. 1).

Upon treating a solution of DTC derived from methoxy styrene (R = -PhOMe) and bis(butylsulfanyl-thiocarbonyl) disulfide (BBTD) in a 1:5 molar ratio with FcPF$_6$ in anhydrous DCM, the corresponding TTC was obtained with an 90% yield (Table 2, Entry 1). Simultaneously, TTC-DTC was also generated. Interestingly, TTCs derived from styrene were found to effectively control further radical SUMI processes. However, DTCs originating from styrene, acrylate, and acrylonitrile failed to convert to TTCs under standard conditions due to the instability of

carbocations generated by the heterolysis of DTCs (Table 2, Entries 2 and 3). In the case of vinylether-derived DTC, the conversion to TTC occurred but with a low yield of 23% under standard conditions (Table 2, Entry 4). Exploration of the impact of reaction time on the process revealed a progressive increase in yield before reaching 24 h, after which there was no substantial increase due to the reversible nature of the reaction (Supplementary Fig. 31). Upon analyzing the HPLC curve of the reaction over a period of 36 h, it was observed that, apart from the unreacted DTC (~3% of the total), the catalyst, and the desired products, there were ~3% impurities present (Supplementary Fig. 31). After conducting additional purification steps for 6, 12, and 36 h, the product exhibited yields of 52%, 75%, and 91%, respectively, which closely correlated with their respective conversion rates (Table 2, Entries 5–7).

Reducing the amount of BBTD also resulted in reduced yields, with thiocarbamate yields ranging from 46% to 70% when 1–2 equivalents of BBTD were used (Table 2, Entries 8 and 9). No thiocarbamate was produced in the absence of FcPF$_6$ (Table 2, Entry 10). Introducing photoredox via 2,4,6-tris(p-methoxyphenyl)pyrylium tetrafluoroborateas (TMPP) under 460 nm led to a 60% yield of thiocarbamate (Table 2, Entry 11). Interestingly, ZnTPP was unable to activate DTC to generate radicals via the PET process, resulting in no reaction under irradiation in a MeCN solution of DTC, BBTD, and ZnTPP (Table 2, Entry 12). The photoiniferter method was used to convert DTC to TTC, resulting in a 35% yield of TTCs by irradiating the acetonitrile solution of DTC and BBTD directly with UV light (365 nm). The low yield can be attributed to the ability of UV light to mediate the production of radicals from TTCs (Table 2, Entry 13).

This DTC-to-TTC reaction was then utilized for a mechanism transformation from cationic SUMI to radical SUMI (Fig. 3a). TTCs (12-DTC to 17-DTC) were obtained from the one- and two-step cationic SUMI of vinyl ethers and styrenes into DTCs, as reported in our recent method[30] (Supplementary Figs. 32–35). These DTCs were then exchanged with BBTD under the initiation of FcPF$_6$ to obtain the corresponding TTCs (12-TTC to 16-TTC, and Supplementary Figs. 36–41) with yields ranging from 75% to 92%. Bis(dodecylsulfanyl-thiocarbonyl) disulfide (BDTD) with longer alkyl chains also reacted with 17-DTC under standard conditions, providing an 84% yield for 17-TTC (Supplementary Fig. 42). Importantly, alkynyl and ester functional groups remained stable under exchange reaction conditions. Additionally, the exchange reactions of oligomer DTCs were efficiently processed.

Subsequently, further radical SUMI of suitable monomers into TTCs prepared by the integration of cationic SUMI and thiocarbonylthio exchange reactions was explored (Fig. 3b). In toluene, AIBN initiated the SUMI process of oligomer TTCs (12-TTC and 17-TTC) and maleimides in a 1:1 molar ratio, generating 18-TTC and 19-TTC with yields of 92–88%, respectively. The second-step radical SUMI was then investigated by inserting vinyl ether into 19-TTC to obtain a mono-adduct, 20-TTC, with a yield of 82% (Supplementary Figs. 43–46). These results highlight the successful transformation mechanism from cationic SUMI to radical SUMI through FcPF$_6$-initiated DTC-to-TTC reactions, allowing for multi-step carbon-carbon single bond formation among various monomers via this strategy.

## Sequential radical-cationic-radical SUMIs

One of the major challenges in the synthesis of discrete vinyl oligomers via SUMI is the preparation of longer oligomers with controllable structure. Because efficient SUMI requires the selection of a suitable monomer/CTA pair, the choice of monomers for subsequent SUMI is reduced. Xu, Boyer, and coworkers reported a solution for the preparation of discrete oligomers with theoretical "infinite" chain extension by sequential and alternating SUMI processes[21].

Our approach builds on the concept of mechanistic alternating SUMI, wherein the oligomer chain is grown by alternating the use of radical and cationic SUMIs. Expanding on the chain length of oligomer

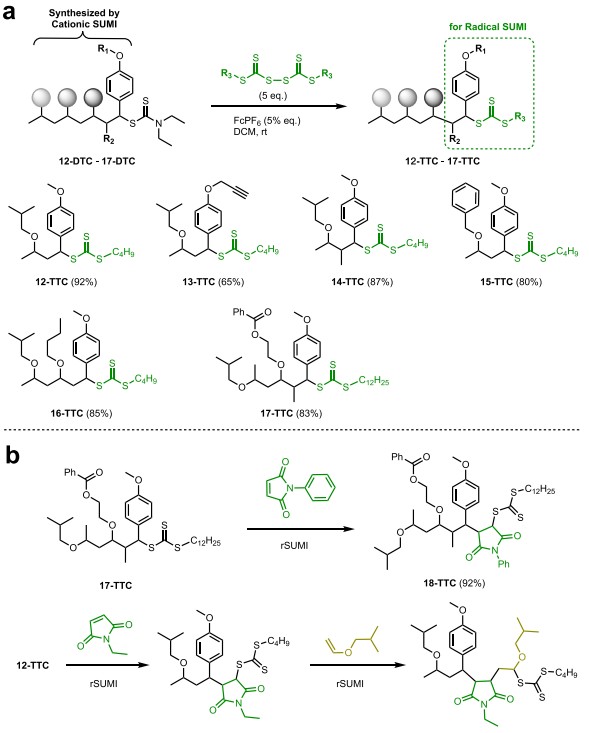

**Fig. 3 | Transformation from dithiocarbamates to trithiocarbonates and further radical SUMI reactions. a** Transformation from dithiocarbamates (DTCs) to trithiocarbonates (TTCs) via exchange reaction with bis(sulfanyl-thiocarbonyl) disulfide. Optimizations were performed using DTC (12-DTC − 17-DTC, 1 eq.), BBTD/BDTD (5 eq.), and FcPF$_6$ (1 mol %) in dichloromethane over a period of 24 h. **b** Following radical SUMI of maleimides or vinyl ethers. Optimizations were performed using TTC (12-TTC − 17-TTC,1 eq.), maleimide (1.2 eq.), and AIBN (5 mol %) in toluene under 70 °C over a period of 24 h. FcPF$_6$ = ferrocenium hexafluorophosphate, BBTD = bis(butylsulfanyl-thiocarbonyl) disulfide, BDTD = bis(dodecylsulfanyl-thiocarbonyl) disulfide, AIBN = 2,2'-azobis(2-methylpropionitrile).

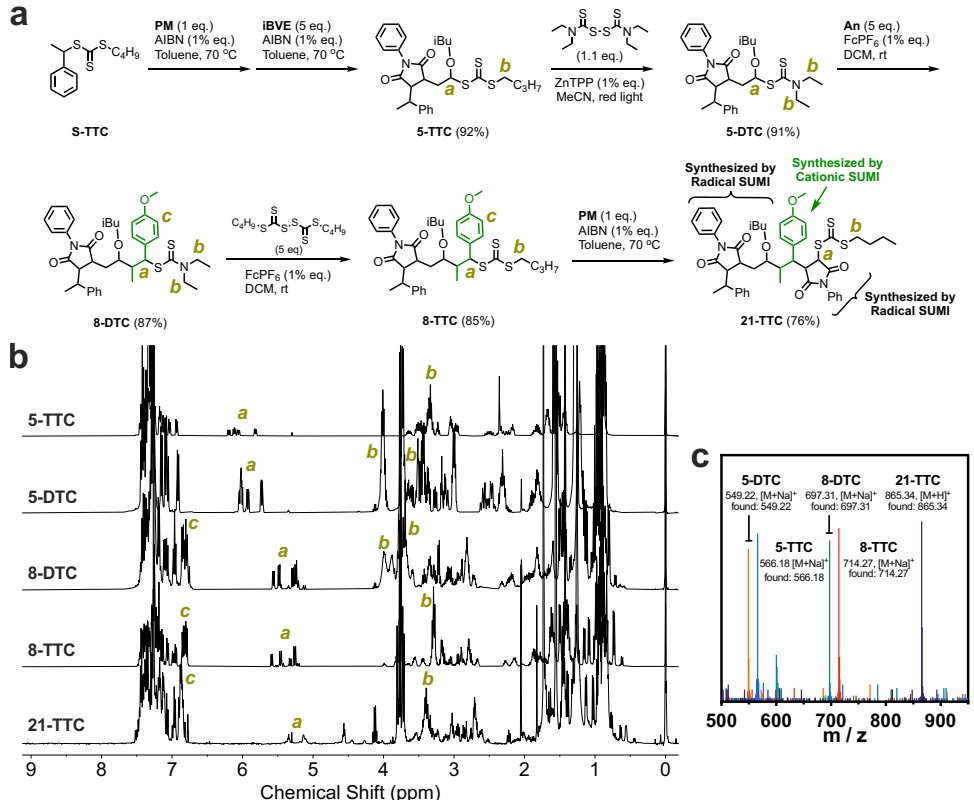

**Fig. 4 | Sequential radical-cationic-radical SUMIs. a** Synthesis route for discrete tetramer (21-TTC) by sequential radical-cationic-radical SUMIs. **b** $^{1}$H NMR spectra in CDCl$_3$ and **c** ESI mass spectra for 21-TTC and precursors.

8-DTC, synthesized through the prior radical SUMI and subsequent cationic SUMI processes (Fig. 4a), we conducted further chain extension. 8-DTC underwent thiocarbonylthio transfer by bis(butylsulfanyl-thiocarbonyl) disulfide (BBTD) to yield 8-TTC with a 85% yield under standard conditions. The subsequent addition to *N*-phenyl maleimide (PM) via AIBN-initiated radical SUMI became feasible, resulting in the formation of 21-TTC with a 76% yield. Through the sequential radical-cationic-radical SUMI with a six-step synthesis process, four new carbon-carbon single bonds were generated, achieving an overall yield of 37.7% (Fig. 4b, c).

### Interconvertible radical and cationic RAFT polymerization via switching thiocarbonylthio end-groups

In addition to facilitating the controlled conversion of radical and cationic SUMIs, the alteration of thiocarbonylthio end-groups can also be applied to achieve the transformation of radical and cationic RAFT polymerization. We then endeavor to produce well-defined block copolymers incorporating cationic polymerizable monomers like iso-butylvinyl ether (iBVE) and 4-methoxystyrene (MOS), alongside radical polymerizable monomers such as methyl acrylate (MA) and styrene (S). This is achieved through oriented end-group transformation, coupled with both cationic and radical RAFT polymerization processes.

In the initial case study, we investigate the transition from cationic to radical RAFT polymerization (Fig. 5a and Supplementary Fig. 47). When employing PiBVE-DTC with an $M_n$ of 2060 (GPC) and a degree of polymerization (DP) of 20 ($^{1}$H NMR), obtained through cationic RAFT polymerization of iBVE (Supplementary Fig. 48), direct control of radical polymerization of MA initiation by AIBN resulted in a polymer with a bimodal GPC curve (Supplementary Fig. 50). This phenomenon is attributed to the unsuitability of the thiocarbonylthio end (Z group) with lower reactivity toward PMA• radical addition, as well as the

ineffective homolytic leaving group and initiating radical of the R group.

The primary solution to this issue involves transforming both R and Z groups to make them suitable for the radical RAFT polymerization of conjugated vinyl monomers (CVMs). The transformation of the R group entails changing it from a vinyl-ether derivative to a styrene derivative through the cationic SUMI method, following the procedure detailed in our previous literature[30]. Treatment of PiBVE-DTC and MOS with FcPF$_6$ in anhydrous DCM yielded PiBVE-MOS-DTC (Fig. 2b, c and Supplementary Fig. 49). Subsequent transformation of the Z group converts the macro-dithiocarbonate PiBVE-MOS-DTC into the trithiocarbonate, rendering it suitable for the radical RAFT polymerization of CVMs such as MA or styrene. The reaction of PiBVE-MOS-DTC and BDTD with FcPF$_6$ produced the trithiocarbonate-end PiBVE-MOS-TTC. The GPC analysis displayed a unimodal distribution ($M_w/M_n = 1.05$) that slightly shifted towards higher molecular weight compared to PiBVE-MOS-DTC (Fig. 5c). Additionally, the matrix-assisted laser desorption/ionization-time of flight (MALDI-TOF) mass spectra revealed a single set of peaks at intervals of 100 Da, representing the iBVE repeating units, with only a few peaks (<5%) showing the absence of thiocarbonylthio end groups (Supplementary Fig. 51). Each peak exhibited an increase in molecular weight by 129 Da relative to PiBVE-MOS-DTC, indicating the substitution of DTC with TTC (Fig. 5b). Finally, PiBVE-*b*-PMA and PiBVE-*b*-PS block copolymers were successfully prepared by utilizing PiBVE-MOS-TTC as a macro-RAFT agent under radical polymerization conditions. The GPC traces exhibited a unimodal shape, shifting to higher molecular weight for both PiBVE-*b*-PMA and PiBVE-*b*-PS block copolymers (Fig. 5c).

In the second instance, the initial PMA block was synthesized through radical RAFT polymerization, succeeded by the creation of a PiBVE or PMOS block through cationic RAFT (Fig. 5d and Supplementary Fig. 52). Attempts at directly synthesizing PMA-*b*-PiBVE

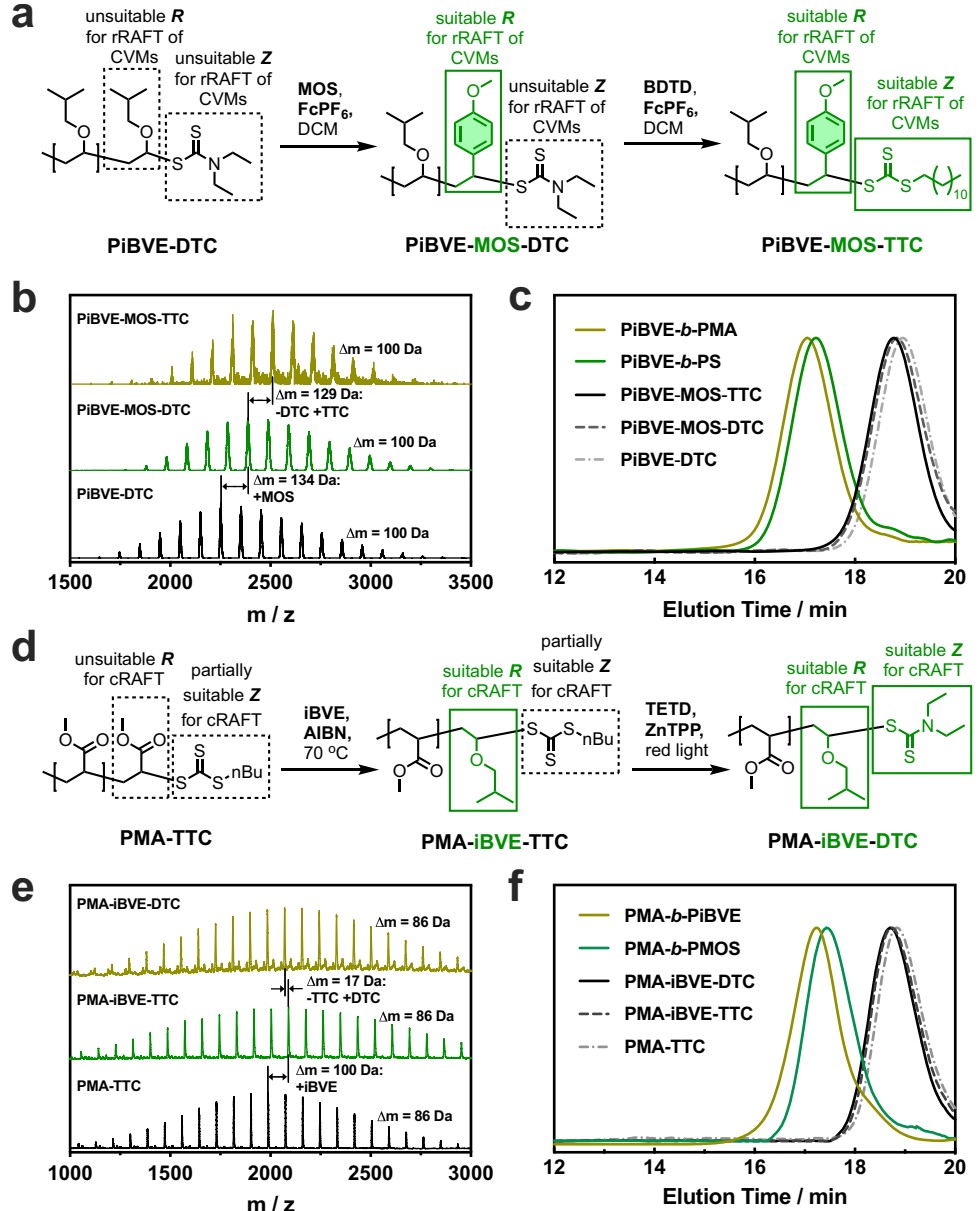

**Fig. 5 | Interconvertible radical and cationic RAFT polymerization via switching thiocarbonylthio end-groups. a** Synthesis of PiBVE-*b*-PMA and PiBVE-*b*-PS block copolymers through the transition from cationic to radical RAFT polymerization by DTC-to-TTC switching. CVM conjugated vinyl monomer. **b** MALDI-TOF MS spectra and **c** GPC trances for PiBVE precursor, intermediate, and block copolymers. **d** Synthesis of PMA-*b*-PiBVE and PMA-*b*-PMOS through the transition from radical to cationic RAFT polymerization by TTC-to-DTC switching. **e** MALDI-TOF MS spectra and **f** GPC trances for PMA precursor, intermediate, and block copolymers.

through cationic RAFT polymerization of iBVE starting from a PMA-TTC macro-RAFT yielded no significant yield of the block copolymer (Supplementary Figs. 52 and 55). The electron-withdrawing R group of PMA-TTC proved incompatible with cationic fragmentation. Subsequently, the R group derived from MA was transferred to the cationic RAFT-suitable moiety through radical SUMI between PMA-TTC and iBVE (Fig. 5d, e and Supplementary Fig. 54). This macroRAFT agent subsequently underwent a Z-transfer to generate a cationic RAFT-suitable dithiocarbamate by reacting with TETD and ZnTPP under red light irradiation. This reaction led to the production of PMA-iBVE-DTC following separation achieved via column chromatography. The GPC analysis revealed a single-peaked distribution ($M_w/M_n = 1.07$, Fig. 5e). Moreover, MALDI-TOF MS primarily exhibited a series of peaks (~92% of the total) with an 86 Da interval, representing MA repeat units. Each peak's molecular weight, relative to the PMA-iBVE-TTC peak, was reduced by 17 Da, which corresponded to the substitution of TTC with

DTC (Fig. 5d). It also indicated the presence of a small quantity of unreacted PMA-iBVE-TTC (~4% of the total) and the polymer peak without the thiocarbonylthio end group when subjected to MS conditions (Supplementary Fig. 56). Consequently, PMA-*b*-PiBVE and PMA-*b*-PMOS block polymers were successfully prepared through cationic RAFT polymerization, exhibiting a narrow molecular weight distribution ($M_w/M_n$ of 1.15-1.19, Fig. 5f). These findings underscore that the dual switching of R- and Z-groups enabled the synthesis of well-defined block copolymers containing cationically and radically polymerizable monomers by combining cationic and radical RAFT polymerization.

## Methods
### General procedure for radical SUMI
In a nitrogen filled glove box, trithiocarbonate (1 equiv), monomer (1–5 equiv), AIBN (0.05 equiv), and toluene were charged into an oven-dried 20 mL Schlenk tube equipped with a magnetic stirring bar. After being

stirred for 12–24 h at 70 °C, the crude product was purified by flash chromatography using petroleum ether/ethyl acetate as the eluent.

### General procedure for cationic SUMI

In a nitrogen filled glove box, dithiocarbamates (1 equiv), monomer (1–5 equiv), FcPF$_6$ (0.01 equiv), and DCM were charged into an oven-dried 20 mL Schlenk tube equipped with a magnetic stirring bar. After being stirred for 12–24 h at 25 °C, the crude product was further purified by flash chromatography using PE/EA as the eluent.

### General procedure for converting trithiocarbonate to dithiocarbamate

In a nitrogen filled glove box, trithiocarbonate (1 equiv), TETD (1.1 equiv), ZnTPP (0.01 equiv), MeCN were charged into an oven-dried 20 mL Schlenk tube equipped with a magnetic stirring bar. The mixture was stirred under red light irradiation (630 nm) at room temperature for 6–36 h. The crude product was purified by flash chromatography using petroleum ether/ethyl acetate as the eluent.

### General procedure for converting dithiocarbamate to trithiocarbonate

In a nitrogen filled glove box, dithiocarbamate (1 equiv), BBTD/BDTD (5 equiv), FcPF$_6$ (0.01 equiv), and DCM were charged into an oven-dried 20 mL Schlenk tube equipped with a magnetic stirring bar. The mixture was stirred at room temperature for 6–36 h. The crude product was purified by flash chromatography using petroleum ether/ethyl acetate as the eluent.

## Data availability

The authors declare that all data supporting the findings of this study are available within the article, the associated source data and its Supplementary Information and can also be obtained from the corresponding author. Source data are provided with this paper.

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

## Acknowledgements

We are grateful to the National Natural Science Foundation of China (22334005, R.Y., and 22271089, G.L.) and the Natural Science Foundation of Hunan Province (2024JJ2043, G.L.) for financial support.

## Author contributions

W.H. and W.T. contributed equally. G.L., W.T., and W.H. conceived the idea. W.H, W.T., Z.W., X.L., and G.T. designed and performed the experiment. G.L., J.T., W.H., J.H. and S.Y. analyzed data. G.L. and R.Y. supervised the study. G.L. and W.H. wrote the manuscript.

## Competing interests

The authors declare no competing interests.
