## [Peer Review File · Nature Communications]

Controlled Switching Thiocarbonylthio End-Groups Enables Interconvertible Radical and Cationic Single-Unit Monomer Insertions and RAFT PolymerizationsReviewers' Comments:

Reviewer #1:

Remarks to the Author:

Yang and coworkers reported a RAFT end-group interchange between trithiocarbonate and dithiocarbamate through the exchange reaction of R-SCSZ with corresponding precursors of the other. The compounds before and after the interchange are used to further single unit monomer insertion reactions or polymerization via radical and cationic mechanisms, respectively, or vice versa. However, the paper's readability is compromised by numerous typo errors and confusing or misleading information, as detailed below:

1. The primary novelty of the paper lies in the interchange reaction between trithiocarbonate and dithiocarbamate, showcased in Schemes 2A and 3A, yielding diverse SUMI-adduct structures with varying yields. However, the rationale behind extending different functional monomers to the same R terminal DTC in Scheme 2B, previously demonstrated by the authors in their earlier paper (Ref 29), is unclear. The concept of cationic SUMI no longer constitutes an innovation in this paper; rather, it serves solely to demonstrate the success of the interchange reaction, without adding substantial value. Therefore, the inclusion of one or two characteristic monomers may suffice. Scheme 3B presents a similar issue; the addition of radical SUMI of maleimides with different functional N-substituents does not contribute significantly to the field, as its efficacy has previously been established. Instead, it unnecessarily inflates the supplementary document to 95 pages.
2. A notable observation arises from Table 1 and Scheme 2A. At Line 152, a typographical error is identified: '...certain oligomers such as 12-DTC and 14-DTC....' should read as '6-DTC and 7-DTC.' The synthetic conditions for these two compounds (2 eq disulfiram) differ from 1-DTC (1.1 eq disulfiram) to enhance yields. How much has been improved if compare to 1.1eq? However, in Table 1 #8 and 9, the authors have demonstrated higher ratios of disulfiram produced lower yields. This is confusing. All these three compounds possess the same R terminal group and Z group. The authors attribute this discrepancy to molecular weight differences, which seems implausible given their marginal variance of only 100 D. If correct, this explanation would suggest a significantly lower RAFT interchange yield than 90% in Figure 1 because the molecular weight of the polymers is 2500 D, which is obviously contrary to observed outcomes.
3. While selectivity is evident in the transformation from trithiocarbonate to dithiocarbamate, the reverse process from dithiocarbamate to trithiocarbonate, as discussed by the authors, lacks such selectivity. Therefore, this claim may somewhat overstate the concept..
4. The RAFT interchange achieves its highest yields at 92%, as illustrated in Schemes 2A and 3A, as well as Figures S2 and S28. Consequently, the interchange in the polymer cases (Figure 1) should ideally reach a maximum of 92%, leaving the remaining 8% in the system. This discrepancy could potentially be confirmed by MALDI-TOF and GPC analyses. It is imperative for the authors to address this issue comprehensively rather than overlooking it.
5. In Table 1, #5 indicated a yield of 50% at 12 h, which does not match with the conversion (> 75%) depicted in the kinetic data presented in Figure S2. This discrepancy raises questions about the consistency between the two results. Interestingly, entries #6 and #7 at 18 hours and 36 hours respectively closely match the kinetic data in Figure S2. A similar discrepancy is observed in Table 2, where the data in entries #5, #6, and #7 do not align with that presented in Figure S28.

Minor:

6. Line 120-129: The whole paragraph is repeating the last paragraph.
7. Page S92: 'iBVE (0.55 mmol)' should be 'disulfiram (0.55 mmol)'.
8. Figures S2 and S28: Keep consistent for 'TT' and 'DT' in the figure as 'TTC' and 'DTC'.
9. 1.1 eq Disulfiram to TTC is not 'stoichiometric disulfiram'.
10. Scheme 1: 'proposal mechanism' should be 'proposed mechanism'.
11. Scheme S2: very messy scheme. The reaction conditions for 5-TTC synthesis are completely different from those in its experimental protocol. The monomer for EMA-TTC to EMA-MA-TTC should be 'MA' instead of 'PMI'. The synthesis of 6-TTC showed 5 eq IBVE to EMA-MA-TTC in the Scheme, but 2:1 in the protocol. Scheme S3 and its protocols have a complete mess in the monomers, solvents and ratios.

12. For what reasons, the preparation of 2-DTC necessitates the utilization of the AIBN initiation system to achieve an 88% yield? If not using this initiation system, what is the yield? Why?
13. Figure S13: Assignment of proton c is wrong. It is also wrong for the proton b assignment in Figure S14. The same as Figure S19.

Reviewer #2:

Remarks to the Author:

Interesting work is described that merits publication. However, the paper is not well-written.

A reasonable simple method for interconverting dithiocarbamate and trithiocarbonate RAFT agents is described. When the preferred conditioned are used yields are good (but not quantitative). It is not clear whether byproducts are formed or there is incomplete conversion (possibly both). Blue light gives 76% vs 92% yield under the standard conditions. Does this indicate a lower rate of reaction or other things happening. Most would tolerate a slightly lower reaction rate if it meant a product free of expensive catalyst. Or maybe more intense light or a higher reaction temperature could be used.

I would suggest that more work is required to characterize the process before publication.

In the literature term "SUMI" does not necessarily imply a RAFT mechanism. Although it becomes obvious, the exclusive use the term in this paper to mean radical or cationic RAFT-SUMI may create confusion.

92% may be a good yield, but it is not clear why a 92% yield should be considered impressive. The authors should avoid unnecessary superlatives and allow the reader to judge.

The experimental procedure in the SI should be linked to the details reported in the manuscript.

It is important to also know the actual yield (not just the isolated yield of the intended product). Were by-products formed? Or were corresponding amounts of unchanged starting materials seen. The Figure S2 might contain this information but it is too small, not quantitative and the disulfides are not seen under the conditions. Were the unidentified peaks in the traces identified. Is IBVE-TT initially converted to something else? What were the conditions used for HPLC (the instrument is described but there are no other details).

What form of calibration was used for GPC molar mass determination? Provide more details of GPC analysis.

Attention should be paid to the nomenclature of the disulfides. In various places we have bis(sulfanyl-thiocarbonyl) disulfide, bis(sulfuryl-thiocarbonyl) disulphides, and bistrithiocarbonate disulfide [all of which may be the same thing – but not sure any of the are correct]. Disulfurim is not a commonly encountered name in the field – thiuram disulfide is more common (but still not IUPAC recommended).

Was stereoselective addition observed as in ref 21. The authors should comment.

Reviewer #3:

Remarks to the Author:

The authors propose an innovative technique for the interconversion of radical and cationic single-unit monomer insertion, enabling precise monomer integration within the polymer chain. This development of new techniques for controlling monomer sequence holds significant implications in polymer science and related fields such as nanomedicine, energy, and advanced materials. The key innovation in this

study lies in the efficient switching from cationic SUMI to radical SUMI, facilitating the creation of new sequence-defined polymers. While radical SUMI has been previously explored by Moad, Zard, You, Boyer, and others, the focus has predominantly been on radical SUMI, with limited success in efficiently transitioning to cationic SUMI. This approach offers efficiency and scalability, promising advancements in polymer synthesis and beyond. The paper is very well written, and the data support the conclusions of this study.

In the introduction, the authors should include this work from Boyer's group in collaboration with Hawker and Moad: RAFT-mediated, visible light-initiated single unit monomer insertion and its application in the synthesis of sequence-defined polymers *Polymer Chemistry* 8 (32), 4637-4643.

In figure 1, the authors should add n for the brackets to show that they used a polymer.

A reference should be added to mention previous works on the use of ZnTPP to perform PET (RAFT) and activated trithiocarbonate (*Angewandte Chemie International Edition* 57 (6), 1557-1562).

On page 8, line 90, The concept of selective activation was reported originally in Exploiting metalloporphyrins for selective living radical polymerization tunable over visible wavelengths; *Journal of the American Chemical Society* 137 (28), 9174-9185. A more recent paper from the same group also detailed this selectivity: Selective photoactivation of trithiocarbonates mediated by metal naphthalocyanines and overcoming activation barriers using thermal energy; *Journal of the American Chemical Society* 144 (2), 995-1005

At the end of this sentence, line 99: "facilitates the production of dithiocarbamate (R-DTC) via chain transfer cycles", Scheme 1 should be referenced.

During the exchange of TTC with DTC, the authors used ZnTPP (up to 5%), which is significant. It will be interesting if the authors can comment on the need or not to remove ZnTPP before cationic polymerization.

Line 249, the authors refer to Scheme 7, please check.

The authors have placed a huge amount of data in SI, which is great. However, I think moving such as key NMR results will be interesting in the main text.

Please check the following sentence: "General procedure for transfer from trithiocarbonate to dithiocarbamate" and please check sentence 324.

Thank you very much for reviewing our manuscript (NCOMMS-24-02980-T) entitled “*Interconvertible Radical and Cationic Single-Unit Monomer Insertions via Oriented Switching Thiocarbonylthio End-Groups*”. We have carefully revised the manuscript according to reviewers’ comments. All changes in the revised text were marked. Point-by-point responses to the reviewers’ comments are as follows.

Reviewer #1:

Yang and coworkers reported a RAFT end-group interchange between trithiocarbonate and dithiocarbamate through the exchange reaction of R-SCSZ with corresponding precursors of the other. The compounds before and after the interchange are used to further single unit monomer insertion reactions or polymerization via radical and cationic mechanisms, respectively, or vice versa. However, the paper's readability is compromised by numerous typo errors and confusing or misleading information, as detailed below:

1. The primary novelty of the paper lies in the interchange reaction between trithiocarbonate and dithiocarbamate, showcased in Schemes 2A and 3A, yielding diverse SUMI-adduct structures with varying yields. However, the rationale behind extending different functional monomers to the same R terminal DTC in Scheme 2B, previously demonstrated by the authors in their earlier paper (Ref 29), is unclear. The concept of cationic SUMI no longer constitutes an innovation in this paper; rather, it serves solely to demonstrate the success of the interchange reaction, without adding substantial value. Therefore, the inclusion of one or two characteristic monomers may suffice. Scheme 3B presents a similar issue; the addition of radical SUMI of maleimides with different functional N-substituents does not contribute significantly to the field, as its efficacy has previously been established. Instead, it unnecessarily inflates the supplementary document to 95 pages.

Response: We appreciate the reviewer's suggestion. Indeed, the core innovation of this paper lies in the interchange of Thiocarbonylthio end-groups. However, our primary objective is to design a controlled sequence of vinyl monomers containing monomers suitable for both cationic and radical SUMI reactions. In order to achieve this objective, we have conducted SUMI reactions using various monomers after transforming the thiocarbonylthio end-groups. Although these reactions have been

previously reported in our own and others' literature, we believe they are still relevant and meaningful.

Furthermore, we have taken the reviewer's suggestions into account. We have excluded the synthesis process of certain compounds that have already been reported. Additionally, we have relocated some data from the SI section to the main body, resulting in a more streamlined and organized SI section. As a result, the SI section has been condensed to a total of 71 pages.

2. A notable observation arises from Table 1 and Scheme 2A. At Line 152, a typographical error is identified: '...certain oligomers such as 12-DTC and 14-DTC....' should read as '6-DTC and 7-DTC.' The synthetic conditions for these two compounds (2 eq disulfiram) differ from 1-DTC (1.1 eq disulfiram) to enhance yields. How much has been improved if compare to 1.1eq? However, in Table 1 #8 and 9, the authors have demonstrated higher ratios of disulfiram produced lower yields. This is confusing. All these three compounds possess the same R terminal group and Z group. The authors attribute this discrepancy to molecular weight differences, which seems implausible given their marginal variance of only 100 D. If correct, this explanation would suggest a significantly lower RAFT interchange yield than 90% in Figure 1 because the molecular weight of the polymers is 2500 D, which is obviously contrary to observed outcomes.

Response: We appreciate the reviewer's insightful discussion. We attempted to synthesize these two products using standard techniques, but found that the reaction rate was too slow, with yields of 67% for 6-DTC and 75% for 7-DTC respectively over a longer 48-hour period. Through experimentation, we discovered that increasing the TETD equivalent could speed up the reaction rate and improve the yield, so we used this condition. As for whether this phenomenon is driven by high molecular weight, we concur with the reviewer that molecular weight isn't the main factor, and we have eliminated this discussion in the revised version. At present, we don't have conclusive data regarding why this occurs. We suspect it could be due to penultimate effect or steric hindrance. The following is our revised text.

“Compounds 6-DTC and 7-DTC have lower reaction rates under standard conditions, with yields of 67% and 75% after 48 h, respectively. To enhance the synthesis efficiency, a higher yield can be obtained by using 2 eq. of TETD.”

3. While selectivity is evident in the transformation from trithiocarbonate to dithiocarbamate, the

reverse process from dithiocarbamate to trithiocarbonate, as discussed by the authors, lacks such selectivity. Therefore, this claim may somewhat overstate the concept.

Response: We appreciate your insightful observation. The reviewer accurately noted that the transformation from trithiocarboxylate to dithiocarboxylate exhibits clear selectivity. As we have discussed, the reverse procedure unfortunately does not demonstrate a high level of selectivity. Given the absence of better reaction conditions at this point, our method of enhancing the BBTD equivalent to attain a higher yield is a pragmatic strategy. This aligns with our concept of oriented thiocarbonylthio end group transitions in this study, ensuring a straightforward and efficient interconversion between TTC and DTC.

4. The RAFT interchange achieves its highest yields at 92%, as illustrated in Schemes 2A and 3A, as well as Figures S2 and S28. Consequently, the interchange in the polymer cases (Figure 1) should ideally reach a maximum of 92%, leaving the remaining 8% in the system. This discrepancy could potentially be confirmed by MALDI-TOF and GPC analyses. It is imperative for the authors to address this issue comprehensively rather than overlooking it.

Response: Thanks to the suggestion from the reviewer, we have thoroughly examined the process of the thiocarbonylthio switching reaction of small molecules and polymers under standard conditions, as well as the final composition of the reaction system. The conversion of small molecules TTC to DTC was monitored using HPLC. After 24 h of light exposure, the average conversion rate of TTC raw materials reached ~92%. During this time, the HPLC curve indicated that the impurity content, excluding the product, catalyst, and unreacted raw material, was less than 1%. Furthermore, prolonging the reaction time did not lead to an increase in the conversion rate. These findings suggest that the reversibility of the reaction prevents its quantification, and the impurity content is minimal. We re-separated the system with different reaction time, and the yield of the product was in accordance with the conversion rate of the raw material. We have included this discussion in the revised manuscript.

“We proceeded to monitor the reaction process under standard conditions using high performance liquid chromatography (HPLC, Figure S2). The reaction showed an initial induction period of approximately 2 h, after which it progressed rapidly. Following a 24-hour reaction period, we observed that the conversion of **iBVE-TTC** reached 92% and the HPLC curve did not exhibit significant impurity peaks, except for starting material (**iBVE-TTC** and disulfiram). Furthermore, even upon extending the reaction time to 36 h, there was no increase in the conversion of **iBVE-TTC**. This suggests that the inability to achieve quantitative conversion is attributed to the reversible nature of the reaction. We further purified the reaction systems for 8, 12, and 36 h, yielding 36%, 74%, and 91%, respectively, which closely match their conversion. (Table 1, entries 5-7).”

Similar results were obtained for the conversion of small molecule DTC to TTC. The using of 5 equivalent BBTC slightly enhanced the conversion rate of MOS-DTC; however, it also resulted in the formation of a small number of by-products (~3 %). We re-separated the system with different reaction time, and the yield of the product was in accordance with the conversion rate of the raw material. This discussion has been incorporated into the revised manuscript.

“Exploration of the impact of reaction time on the process revealed a progressive increase in yield before reaching 24 hours, after which there was no substantial increase due to the reversible nature of the reaction (Figure S26). Upon analyzing the HPLC curve of the reaction over a period of 36 h, it was observed that, apart from the unreacted DTC (~3% of the total), the catalyst, and the desired products, there were ~3% impurities present (Figure S26c). After conducting additional purification steps for 6, 12, and 36 hours, the product exhibited yields of 52%, 75%, and 91%, respectively, which closely correlated with their respective conversion rates (Table 2, Entries 5-7).”

We also examined the effectiveness of converting polymer thioester end groups. By employing GPC characterization, we observed that the converted polymer displayed a symmetrical single-mode peak. However, due to the minor change in molecular weight resulting from the conversion process, the information obtained from the GPC curve was limited. MALDI-TOF analysis yielded more insightful results. In the case of TTC to DTC conversion, we achieved a conversion rate of 98%, with only a few signals indicating the removal of thiocarbonylthio end group under MS conditions. This indicates that over 95% of the polymers were successfully transformed into DTC-containing polymers. When converting DTC to TTC, the conversion rate of DTC approached 93%, but a small

amount of unconverted DTC and products without thiocarbonylthio end groups were detected. These discussions have been incorporated into the revised manuscript.

“The GPC analysis displayed a unimodal distribution ($M_w/M_n = 1.05$) that slightly shifted towards higher molecular weight compared to **PiBVE-MOS-DTC** (Figures 2C). Additionally, the matrix-assisted laser desorption/ionization-time of flight (MALDI-TOF) mass spectra revealed a single set of peaks at intervals of 100 Da, representing the *iBVE* repeating units, with only a few peaks (< 5%) showing the absence of thiocarbonylthio end groups (Figure S46). Each peak exhibited an increase in molecular weight by 129 Da relative to **PiBVE-MOS-DTC**, indicating the substitution of DTC with TTC (Figures 2B).”

“The GPC analysis revealed a single-peaked distribution ($M_w/M_n = 1.07$, Figures 2E). Moreover, MALDI-TOF MS primarily exhibited a series of peaks (~92% of the total) with an 86 Da interval, representing MA repeat units. Each peak's molecular weight, relative to the **PMA-iBVE-TTC** peak, was reduced by 17 Da, which corresponded to the substitution of TTC with DTC (Figure 2D). It also indicated the presence of a small quantity of unreacted **PMA-iBVE-TTC** (~4% of the total) and the polymer peak without the thiocarbonylthio end group when subjected to MS conditions (Figure S50).”

5. In Table 1, #5 indicated a yield of 50% at 12 h, which does not match with the conversion (> 75%) depicted in the kinetic data presented in Figure S2. This discrepancy raises questions about the consistency between the two results. Interestingly, entries #6 and #7 at 18 hours and 36 hours respectively closely match the kinetic data in Figure S2. A similar discrepancy is observed in Table 2, where the data in entries #5, #6, and #7 do not align with that presented in Figure S28.

Response: We have carried out the separation and purification of the reaction system again. For detailed results, please refer to the response provided in Question 4.

Minor:

6. Line 120-129: The whole paragraph is repeating the last paragraph.

Response: In the revised version, this error has been rectified.

7. Page S92: 'iBVE (0.55 mmol)' should be 'disulfiram (0.55 mmol)'.

Response: In the revised version, this error has been rectified.

8. Figures S2 and S28: Keep consistent for 'TT' and 'DT' in the figure as 'TTC' and 'DTC'.

Response: In the revised version, this error has been rectified.

9. 1.1 eq Disulfiram to TTC is not 'stoichiometric disulfiram'.

Response: In the revised version, this error has been rectified.

10. Scheme 1: 'proposal mechanism' should be 'proposed mechanism'.

Response: In the revised version, this error has been rectified.

11. Scheme S2: very messy scheme. The reaction conditions for 5-TTC synthesis are completely different from those in its experimental protocol. The monomer for EMA-TTC to EMA-MA-TTC should be 'MA' instead of 'PMI'. The synthesis of 6-TTC showed 5 eq IBVE to EMA-MA-TTC in the Scheme, but 2:1 in the protocol. Scheme S3 and its protocols have a complete mess in the monomers, solvents and ratios.

Response: In the revised version, this error has been rectified.

12. For what reasons, the preparation of 2-DTC necessitates the utilization of the AIBN initiation system to achieve an 88% yield? If not using this initiation system, what is the yield? Why?

Response: Our apologies for the incorrect labeling. It's actually the AIBN initiation system that is 4-DTC. The primary reasoning behind this is due to the presence of a hydroxyl group in 4-DTC, which might interfere with the photoredox process, leading to lower yields under standard conditions. We have included these discussions in the revised text. *"To prepare 4-DTC with a hydroxyl group, the AIBN initiation system is used to achieve a 75% yield, as the presence of the hydroxyl group may affect the photoredox process."*

13. Figure S13: Assignment of proton c is wrong. It is also wrong for the proton b assignment in Figure S14. The same as Figure S19.

Response: In the revised version, this error has been rectified.

Reviewer #2 (Remarks to the Author):

Interesting work is described that merits publication. However, the paper is not well-written.

1. A reasonable simple method for interconverting dithiocarbamate and trithiocarbonate RAFT agents is described. When the preferred conditioned are used yields are good (but not quantitative). It is not clear whether byproducts are formed or there is incomplete conversion (possibly both).

Response: In the revised version, this matter is thoroughly investigated. For detailed results, please refer to the response provided in Question 4, Reviewer #1.

2. Blue light gives 76% vs 92% yield under the standard conditions. Does this indicate a lower rate of reaction or other things happening. Most would tolerate a slightly lower reaction rate if it meant a product free of expensive catalyst. Or maybe more intense light or a higher reaction temperature could be used. I would suggest that more work is required to characterize the process before publication.

Response: We appreciate the reviewer's assessment. Our objective is to mediate the conversion of TTC to DTC through the photoiniferter method without photocatalyst. However, initiating reaction either by direct blue light irradiation or thermal initiation of AIBN lacks sufficient selectivity for the homolysis of C-S bonds in TTC and DTC, hence preventing an efficient TTC-DTC conversion. We have conducted a study on the reaction process triggered by blue light (Figure S3 in revised SI). The reaction was quick during the initial stage, but equilibrium was attained after 24 hours when forward and reverse reactions paced at equal rates. Enhancing yield proves difficult through extending duration or increasing light intensity.

“Without ZnTPP, blue light irradiation (460 nm) directly initiated the reaction that involved homolytic cleavage of C-S bonds. This led to a relative decrease in DTC yield (76%, Table 1, entry 11). The investigation into the reaction procedure revealed that extending the reaction duration did not enhance the conversion rate (Figure S3). Therefore, the lower yield can be attributed to the fact that blue light also mediates the breakdown of DTC and triggers the reverse transformation of DTC to TTC.”

3. In the literature term "SUMI" does not necessarily imply a RAFT mechanism. Although it becomes obvious, the exclusive use the term in this paper to mean radical or cationic RAFT-SUMI may create confusion.

Response: We appreciate the reviewer's insightful comments. While the term SUMI originated from RAFT polymerization, the SUMI concept encompasses multiple mechanisms. Therefore, the term RAFT-SUMI is a more precise expression for this study. We have addressed this point in the introduction. “Various methods have been reported, including polymerization of oligomers with prearranged monomers, template polymerization, chromatographic separation of polydisperse polymers, atom transfer radical addition (ATRA), aminoxyl-mediated single-unit monomer insertion (NM-SUMI), and reversible-addition-fragmentation chain transfer-SUMI (RAFT-SUMI). The development of RAFT-SUMI stemmed from an in-depth exploration of the early stages of RAFT polymerization, revealing the preferential generation of monoadduct products before further chain

growth.”

4. 92% may be a good yield, but it is not clear why a 92% yield should be considered impressive. The authors should avoid unnecessary superlatives and allow the reader to judge.

Response: We appreciate the feedback. The mentioned statement has been removed in the revised version.

5. The experimental procedure in the SI should be linked to the details reported in the manuscript.

Response: The primary experimental procedures have been incorporated into the revised version.

6. It is important to also know the actual yield (not just the isolated yield of the intended product). Were by-products formed? Or were corresponding amounts of unchanged starting materials seen. The Figure S2 might contain this information but it is too small, not quantitative and the disulfides are not seen under the conditions. Were the unidentified peaks in the traces identified. Is IBVE-TT initially converted to something else? What were the conditions used for HPLC (the instrument is described but there are no other details).

Response: In the revised version, this matter is thoroughly investigated. For detailed results, please refer to the response provided in Question 4, Reviewer #1. HPLC conditions are included in the figure caption of Figure S2 and Figure S26. *“Figure S2. Time-evolution of (a and c) HPLC traces (MeCN/H₂O 8/2 v/v; 300 nm absorbance) and (b) conversion recorded for TTC-to-DTC transformation. Reaction was performed on 10 mmol scale using trithiocarbonate (iBVE-TTC, 1 eq.), disulfiram (1.1 eq.), and ZnTPP (1 mol %) in acetonitrile (MeCN) under red light irradiation (630 nm).”* *“Figure S26. Time-evolution of (a and c) HPLC traces (MeCN/H₂O 75/25 v/v; 300 nm absorbance) and (b) conversion recorded for DTC-to-TTC transformation. Reaction was performed on 10 mmol scale using MOS-DTC (1 eq.), BBTD (5 eq.), FcPF₆ (1 mol %) in DCM.”*

7. What form of calibration was used for GPC molar mass determination? Provide more details of GPC analysis.

Response: This information has been added to the modified SI. *“Molecular weights and molecular weight distributions were determined by GPC using an SSI pump equipped 2x Shodex GPC KD column (separation range of molecular weight from 200,000 to 500 Da) and Wyatt Optilab refractive index detector ($\lambda = 658$ nm, 35 °C). THF was used as eluent at a flow rate of 1.0 mL/min. A series of low polydispersity polystyrene standards were employed for calibration. Raw data were processed with the Astra V software (Wyatt Technology). Matrix-assisted laser*

desorption/ionization-time of flight (MALDI-TOF) mass spectra were acquired on an Autoflex Speed MALDI-TOF mass spectrometer (Bruker Daltonics, Germany) equipped with a Smart beam-II laser (355 nm, 1 kHz, Bruker Daltonics).”

8. Attention should be paid to the nomenclature of the disulfides. In various places we have bis(sulfanyl-thiocarbonyl) disulfide, bis(sulfuryl-thiocarbonyl) disulphides, and bistrithiocarbonate disulfide [all of which may be the same thing – but not sure any of the are correct]. Disulfurim is not a commonly encountered name in the field – thiuram disulfide is more common (but still not IUPAC recommended).

Response: We appreciate your correction, and we have made the necessary changes by replacing all instances of “disulfiram” with “tetraethylthiuram disulfide (TETD)”.

9. Was stereoselective addition observed as in ref 21. The authors should comment.

Response: Xu, Boyer, and colleagues have made substantial contributions to the field of stereoselective SUMI, including the literature that the reviewer referred to. They've focused on cyclic vinyl or 1, 2-disubstituted vinyl monomers. Just like theirs, our study also employs maleimide, a cyclic vinyl monomer. Based on their PET-RAFT synthesis methods, similar outcomes can be expected. However, most other SUMI processes employing single substituted vinyl monomers lack stereoselectivity. Although important, this aspect is not the central focus of this study and is therefore not extensively discussed in the main text.

Reviewer #3 (Remarks to the Author):

The authors propose an innovative technique for the interconversion of radical and cationic single-unit monomer insertion, enabling precise monomer integration within the polymer chain. This development of new techniques for controlling monomer sequence holds significant implications in polymer science and related fields such as nanomedicine, energy, and advanced materials. The key innovation in this study lies in the efficient switching from cationic SUMI to radical SUMI, facilitating the creation of new sequence-defined polymers. While radical SUMI has been previously explored by Moad, Zard, You, Boyer, and others, the focus has predominantly been on radical SUMI, with limited success in efficiently transitioning to cationic SUMI. This approach offers efficiency and scalability, promising advancements in polymer synthesis and beyond. The paper is very well written, and the data support the conclusions of this study.

1. In the introduction, the authors should include this work from Boyer's group in collaboration with Hawker and Moad: RAFT-mediated, visible light-initiated single unit monomer insertion and its application in the synthesis of sequence-defined polymers *Polymer Chemistry* 8 (32), 4637-4643.

Response: This literature is referenced in the updated version (ref. 25).

2. In figure 1, the authors should add n for the brackets to show that they used a polymer.

3. A reference should be added to mention previous works on the use of ZnTPP to perform PET (RAFT) and activated trithiocarbonate (*Angewandte Chemie International Edition* 57 (6), 1557-1562).

Response: This literature is referenced in the updated version (ref. 48).

4. On page 8, line 90, The concept of selective activation was reported originally in Exploiting metalloporphyrins for selective living radical polymerization tunable over visible wavelengths; *Journal of the American Chemical Society* 137 (28), 9174-9185. A more recent paper from the same group also detailed this selectivity: Selective photoactivation of trithiocarbonates mediated by metal naphthalocyanines and overcoming activation barriers using thermal energy; *Journal of the American Chemical Society* 144 (2), 995-1005

Response: This literature is referenced in the updated version (ref. 49).

5. At the end of this sentence, line 99: "facilitates the production of dithiocarbamate (R-DTC) via chain transfer cycles", Scheme 1 should be referenced.

Response: In the revised version, Scheme 1 has been referenced.

6. During the exchange of TTC with DTC, the authors used ZnTPP (up to 5%), which is significant. It will be interesting if the authors can comment on the need or not to remove ZnTPP before cationic polymerization.

Response: Indeed, removing ZnTPP from the system, especially when higher concentrations are used in the polymer system, could be challenging. Nevertheless, as reported by Satoh and colleagues (*Polym. J.* 2020, 52, 65), ZnTPP doesn't significantly interfere with the cationic RAFT polymerization catalyzed by B(C₆F₅)₃. Following ZnTPP usage, PMA-iBVE-DTC samples were repurified via column chromatography until the ZnTPP signal was no longer detectable in the HPLC analysis. Initiating the polymerization of iBVE monomers using these samples yielded similar GPC results as before. So, taking into account both prior report and our results, it appears that residual ZnTPP minimally affects the subsequent cationic polymerization. We modified the

description of the experimental procedure and included the corresponding discussion in the revised manuscript, using the new GPC curve (PMA-b-PiBVE) in Figure 2F. “*This macroRAFT agent subsequently underwent a Z-transfer to generate a cationic RAFT-suitable dithiocarbamate by reacting with TETD and ZnTPP under red light irradiation. This reaction led to the production of PMA-iBVE-DTC following separation achieved via column chromatography.*”

7. Line 249, the authors refer to Scheme 7, please check.

Response: In the revised version, this error has been rectified, and “Scheme 7” has been updated to the accurate “Scheme 3”.

8. The authors have placed a huge amount of data in SI, which is great. However, I think moving such as key NMR results will be interesting in the main text.

Response: Thanks for suggestion. The ^1H NMR and mass spectra of 25-TTC, along with the synthetic precursors, have been shifted from the SI to the main text. They have been merged with the original Scheme 4 to create the new Figure 1.

9. Please check the following sentence: "General procedure for transfer from trithiocarbonate to dithiocarbamate" and please check sentence 324.

Response: In the revised version, this error has been rectified.

Reviewers' Comments:

Reviewer #1:

Remarks to the Author:

The authors did not address properly my comments. See the following details:

Comment 1: The authors claim that the "core innovation" lies in the interchange of Thiocarbonylthio end-groups, yet the primary objective is cationic and radical SUMI for sequence-defined polymer. However, the last part of the Results and Discussion section, titled "Interconvertible Radical and Cationic RAFT Polymerization via Switching Thiocarbonylthio End-Groups," does not align with SUMI. Instead, it focuses solely on the interchange of thiocarbonylthio, which should be the primary focus of this article. Additionally, the inclusion of synthetic compounds using the same family of monomers but with different side functionalities does not contribute to innovation; rather, it serves as a distraction. The reviewer did not observe any changes in Scheme 2 and 3, despite the authors' claim that they excluded the synthesis process of certain compounds that had already been reported. It remains unclear what exactly has been excluded in the revised manuscript.

Comment 2: The authors ignored an important comment regarding Table 1, specifically entries #8 and #9, which demonstrated higher ratios of disulfiram (> 2 eq) resulting in lower yields, a confusing observation with those cases in 6-DTC and 7-DTC. The question is repeated because of this conflicting data: why does 2 eq disulfiram lead to low yields due to reverse conversion from DTC to TTC, but increase the yields in 6-DTC and 7-DTC? Furthermore, why was the reverse conversion from DTC to TTC reduced in these instances? This discrepancy lacks coherence and warrants clarification.

Comment 3: The authors did not address the question from the reviewer regarding the 'selectivity'. A single way 'selectivity' should not be overclaimed as whole system 'selectivity'.

Comment 5: The authors claimed that they "carried out the separation and purification of the reaction system again." However, it is evident that the reaction hours listed in Tables 1 and 2 (#5, #6 and #7) were arbitrarily changed from "12, 18, 36 hrs" in the original tables to "8, 12, and 36 hrs" in the revised version. This suggests a lack of attention to detail regarding the presented data. The apparent mismatch of this crucial information was not identified by the authors themselves. As a result, the reviewer is compelled to criticize the accuracy of the data and the professionalism exhibited by the authors.

Comment 12: The claim that "the presence of the hydroxyl group may affect the photoredox process" appears unsubstantiated and lacks supporting reasons or references.

Reviewer #2:

Remarks to the Author:

The authors have addressed the technical queries raised previously but there are still problems with English. The problems include poor word choice, awkward sentence structure, incorrect tense, and singular vs plural confusion.

"artificial" is the wrong word, there is nothing particularly artificial about vinyl polymers.

"stir magneton" should be magnetic stirring bar.

What the authors call dimer, trimer, tetramer, seem to be unimer, dimer, trimer, respectively. They have decided to call the initial RAFT agent a first unit.

Note that these are examples of errors not a complete list.

line 51. Indicate or provide a reference to the "numerous applications."

Usually, high resolution mass spectra would be provided with more digits of precision.

Reviewer #3:

Remarks to the Author:

In my opinion, the authors have addressed the reviewers' comments.

Dr. Johannes Kreutzer
Chief Editor Bio- and Organic Chemistry
Nature Communications

Dear Dr. Kreutzer:

Thank you very much for reviewing our manuscript (NCOMMS-24-02980A) entitled “*Controlled Switching Thiocarbonylthio End-Groups Enable Interconvertible Radical and Cationic Single-Unit Monomer Insertions/RAFT Polymerizations*”. Based on the suggestions of the reviewers, we have revised the title of our manuscript. We have carefully revised the manuscript according to reviewers’ comments. All changes in the revised text were marked. Point-by-point responses to the reviewers’ comments are as follows.

Reviewer #1 (Remarks to the Author):

The authors did not address properly my comments. See the following details:

Comment 1: The authors claim that the "core innovation" lies in the interchange of Thiocarbonylthio end-groups, yet the primary objective is cationic and radical SUMI for sequence-defined polymer. However, the last part of the Results and Discussion section, titled "Interconvertible Radical and Cationic RAFT Polymerization via Switching Thiocarbonylthio End-Groups," does not align with SUMI. Instead, it focuses solely on the interchange of thiocarbonylthio, which should be the primary focus of this article. Additionally, the inclusion of synthetic compounds using the same family of monomers but with different side functionalities does not contribute to innovation; rather, it serves as a distraction. The reviewer did not observe any changes in Scheme 2 and 3, despite the authors' claim that they excluded the synthesis process of certain compounds that had already been reported. It remains unclear what exactly has been excluded in the revised manuscript.

Response: Thank you for your insightful comments and suggestions. We agree with your assessment that the core innovation of our work lies in the controllable switching of thiocarbonylthio end-groups, which not only facilitates the transformation between RAFT polymerization mechanisms but also the

single-unit monomer insertion (SUMI) processes. Accordingly, we have revised the title to better reflect this focus: "*Controlled Switching Thiocarbonylthio End-Groups Enable Interconvertible Radical and Cationic Single-Unit Monomer Insertions/RAFT Polymerizations*". We have also made adjustments to the abstract and introduction sections to emphasize the innovation and applicability of thiocarbonylthio end-group manipulation. These edits better align the manuscript with the transformative capabilities of this technology in polymer synthesis.

Regarding Schemes 2 and 3, we appreciate your observation on the lack of change. We have now revised these schemes to reduce the number of SUMI reaction examples following end-group transfer, thus highlighting the importance of end-group manipulation more effectively. These changes aim to focus on the innovation rather than the reiteration of known synthetic routes.

We believe these revisions address your concerns and better clarify the contributions of our research. We are grateful for your guidance which has undeniably strengthened our manuscript.

Comment 2: The authors ignored an important comment regarding Table 1, specifically entries #8 and #9, which demonstrated higher ratios of disulfiram (> 2 eq) resulting in lower yields, a confusing observation with those cases in 6-DTC and 7-DTC. The question is repeated because of this conflicting data: why does 2 eq disulfiram lead to low yields due to reverse conversion from DTC to TTC, but

increase the yields in 6-DTC and 7-DTC? Furthermore, why was the reverse conversion from DTC to TTC reduced in these instances? This discrepancy lacks coherence and warrants clarification.

Response: Thank you for your insightful comments. We appreciate the opportunity to address the concerns raised regarding the apparent discrepancy highlighted in Table 1, particularly entries #8 and #9. This issue has indeed posed a significant challenge in our studies. To resolve this, we extensively explored the kinetics of the iBVE-TTC to iBVE-DTC transformation under varying concentrations of TETD. Our findings are intriguing: while an excess of TETD can shorten or even eliminate the induction period, it indeed slows down the reaction rate in the later stages. This nuanced behavior is thoroughly discussed in the revised manuscript.

Figure S3. Conversion for TTC-to-DTC transformation. Reaction was performed on 10 mmol scale using trithiocarbonate (iBVE-TTC, 1 eq.), TETD (1.1-5 eq.), and ZnTPP (1 mol %) in acetonitrile (MeCN) under red light irradiation (630 nm).

Scheme S1. proposed mechanism for TTC-to-DTC transformation in presence of ZnTPP under red light irradiation.

“Interestingly, an excess of TETD resulted in a reduction in product yield (Table 1, entries 8 and 9). To explore this phenomenon, we conducted a study on the reaction kinetics under varying ratios of TETD. Our findings indicate that, in contrast to a 1.1-fold of TETD, reactions with a 5-fold of TETD exhibited virtually no induction period, and displayed a significantly faster initial rate, achieving about 42% conversion within 6 h, which is higher than that observed with 2-fold and 1.1-fold excesses of TETD (Figure S3). However, the rate of conversion decreased substantially in the later stages of the reaction. Based on these observations and the proposed mechanism (Scheme S1), we hypothesize that under the photocatalysis of ZnTPP, both TETD and iBVE-TTC are capable of generating radicals (DTC radicals and iBVE radicals). A higher concentration of TETD facilitates the rapid initial production of a large quantity of radicals, thereby promoting the TTC-to-DTC transformation and eliminating the induction period. In the later stages of the reaction, the presence of a large excess of TETD results in DTC radicals preferentially reacting with TETD rather than with iBVE-TTC, leading to a slowed rate of product conversion.”

Additionally, we have conducted kinetic studies on the synthesis of 7-DTC using 1.1-fold or 2-fold of TETD. Our experiments indeed revealed the presence of an induction period even when using a 1.1-fold of TETD, and achieving high conversion rates required a longer reaction time (93% conversion in 72 hours). On the other hand, the use of a 2-fold of TETD resulted in a faster initial rate, albeit with

slightly lower overall conversion rates. Thus, employing a 2-fold excess of TETD can be considered a compromise to enhance synthetic efficiency, even if it means sacrificing slightly higher yields. The discussion on these findings and their implications is provided in the revised manuscript.

Figure S20. Time-evolution of (a) HPLC traces (MeCN/H₂O 8/2 v/v; 300 nm absorbance) and (b) conversion recorded for TTC-to-DTC transformation. Reaction was performed on 10 mmol scale using trithiocarbonate (iBVE-TTC, 1 eq.), TETD (1.1 or 2 eq.), and ZnTPP (1 mol %) in acetonitrile (MeCN) under red light irradiation (630 nm).

“Compounds **6-DTC** and **7-DTC** have lower reaction rates under standard conditions, with yields of 67% and 75% after 48 h, respectively. In our investigation of the reaction kinetics between **7-TTC** and TETD under standard conditions, we observed a notably induction period of ~6 h. Following this induction period, the reaction proceeded rapidly, reaching an ~80% conversion rate at 36 h and ~93% at 72 h (Figure S20). However, when the amount of TETD was increased to a 2-fold molar excess, the induction period was eliminated, with the reaction achieving about ~87% conversion within 24 h. Extending the reaction time beyond this point did not further increase the conversion rate (Figure S20). Based on the observed kinetics of the reaction, to enhance the synthetic efficiency, we employed a 2-fold of TETD for the synthesis of **6-DTC** and **7-DTC**, achieving favorable yields.”

Comment 3: The authors did not address the question from the reviewer regarding the 'selectivity'. A single way 'selectivity' should not be overclaimed as whole system 'selectivity'.

Response: We appreciate the reviewer's comment regarding the use of the term "selectivity" in our manuscript. We apologize for any confusion caused by our previous wording. Upon careful consideration, we agree that it is more appropriate to describe the complete system as "controllable" rather than "selectivity". Similarly, we will clarify that the transformation from TTC to DTC exhibits "selectivity" in specific cases rather than implying a broad selectivity throughout the entire system. We thank the reviewer for bringing this to our attention, and we will make the necessary revisions to ensure the accuracy and rigor of our language in the revised manuscript.

Comment 5: The authors claimed that they "carried out the separation and purification of the reaction system again." However, it is evident that the reaction hours listed in Tables 1 and 2 (#5, #6 and #7) were arbitrarily changed from "12, 18, 36 hrs" in the original tables to "8, 12, and 36 hrs" in the revised version. This suggests a lack of attention to detail regarding the presented data. The apparent mismatch of this crucial information was not identified by the authors themselves. As a result, the reviewer is compelled to criticize the accuracy of the data and the professionalism exhibited by the authors.

Response: Thank you for your comment requesting further clarification on the data acquisition process related to the reaction times reported in our manuscript. We appreciate the opportunity to explain the methodologies involved and the rationale behind the modifications of reported times in our revised document.

Initially, our research aimed to assess the reaction conversion rates by analyzing the isolated yields at different reaction times. Early on, we employed manual column chromatography for separation, which unfortunately led to significant product loss and lower yields. We acknowledge this early oversight and extend our sincere apologies for the lack of precision in our separation process at this stage.

Recognizing the potential discrepancy between separation yields and actual conversion rates due to purification losses, we later adopted a direct analysis of the reaction mixtures using HPLC without prior treatment. This methodological shift was aimed at obtaining more reliable conversion rate data. Following the constructive feedback from Reviewer 1 and Reviewer 2, we reevaluated our HPLC data for the systems at various reaction times during first round of revisions, which led to results that were well-received by the reviewers. We also conducted separations on these systems using a flash chromatography method, achieving isolated yields that matched the HPLC-measured conversion rates.

The reaction times reported in the revised version (8, 12, and 36 hours) represent the adjusted and optimized reaction durations based on these later experiments, intended to provide clearer and more representative data. **It is crucial to emphasize that these are new results obtained from our experiments during first round of revisions.**

We apologize for any confusion caused by not initially detailing this adjustment in our revised manuscript. We have now included a comprehensive explanation of the changes in methodology and the corresponding results obtained in the revision.

Thank you for your diligent review and for allowing us the chance to clarify these aspects. We trust that these adjustments and the additional details provided will address the concerns raised regarding data accuracy and experimental detail.

Comment 12: The claim that "the presence of the hydroxyl group may affect the photoredox process" appears unsubstantiated and lacks supporting reasons or references.

Response: Thank you for your feedback regarding our claim about the influence of the hydroxyl group on the photoredox process. We apologize for the oversight in not initially citing the relevant literature (ref. 52-54) that supports this assertion. In our revised manuscript, we have now included references that detail the reaction between alcohols and intermediates in photoredox processes, which are well-documented in the field. Additionally, we have revised the phrasing of this claim to more accurately reflect the supported hypothesis: "the presence of the hydroxyl group may react with intermediates in the photoredox process." We appreciate your attention to detail and thank you for allowing us the opportunity to correct this oversight. We hope that these amendments address your concerns and strengthen the manuscript.

Reviewer #2 (Remarks to the Author):

The authors have addressed the technical queries raised previously but there are still problems with English. The problems include poor word choice, awkward sentence structure, incorrect tense, and singular vs plural confusion.

"artificial" is the wrong word, there is nothing particularly artificial about vinyl polymers.

"stir magneton" should be magnetic stirring bar.

What the authors call dimer, trimer, tetramer, seem to be unimer, dimer, trimer, respectively. They have decided to call the initial RAFT agent a first unit.

Note that these are examples of errors not a complete list.

Response: We have addressed all the concerns and made the necessary corrections in the revised manuscript. Furthermore, we have conducted a thorough proofreading of the entire manuscript to ensure precise and accurate expression. We believe that these modifications have significantly improved the overall quality of our work.

line 51. Indicate of provide a reference to the "numerous applications."

Response: We apologize for the oversight in not providing a reference to support the statement about the numerous applications. In the revised manuscript, we have included references (ref. 31-33).

Usually, high resolution mass spectra would be provided with more digits of precision.

Response: In the revised version, we have added high-precision mass spectrometry data to ensure clarity and better serve the reader (Figures S25 and S38)

Reviewer #3 (Remarks to the Author):

In my opinion, the authors have addressed the reviewers' comments.

Response: Thank you for acknowledging our revisions and confirming that we have addressed the reviewers' comments satisfactorily. We appreciate your time and effort in reviewing our manuscript and providing constructive feedback, which has undoubtedly enhanced the quality of our work. We look forward to the publication process and hope our research contributes positively to the field. Thank you once again for your support and guidance.

We would like to take this chance to thank you and referees for editing and reviewing this manuscript. The valuable comments have made a much better presentation of this study. We appreciate your continued attention to this manuscript.

Sincerely,

Prof. Guhuan Liu

Key Laboratory of Chemical Biology & Traditional Chinese Medicine Research, MOE
College of Chemistry and Chemical Engineering, Hunan Normal University,
Changsha, Hunan, 410081, China, E-mail:ghliu@hunnu.edu.cn

Reviewers' Comments:

Reviewer #1:

Remarks to the Author:

It's evident that the title contains grammar errors and lacks clarity and readability. A more appropriate title would be 'Controlled Switching of Thiocarbonylthio End-Groups Enables Interconvertible Radical and Cationic RAFT Single-Unit Monomer Insertions and Polymerizations'.

While the authors have made great efforts to address technical issues and rectify misleading data/information within the manuscript, the novelty highlighted by Reviewer 1 remains unsubstantiated. Merely relying on thiocarbonylthio conversion as the core innovation may not suffice for publication in this journal, which caters to general chemistry. Although the concept of switching is conceptually feasible, the practical limitation of a maximum conversion of 92% per switching cycle renders the method potentially useless. Consequently, I do not recommend its publication in this journal. Other professional journals such as *Macromolecules* or *Polymer Chemistry* may be more appropriate venues for this research.